# Ethical Climate, Intrinsic Motivation, and Affective Commitment: The Impact of Depersonalization

**DOI:** 10.3390/ejihpe15040055

**Published:** 2025-04-07

**Authors:** Santiago-Torner Carlos, Jiménez-Pérez Yirsa, Tarrats-Pons Elisenda

**Affiliations:** 1Department of Economics and Business, Faculty of Business and Communication Studies, University of Vic—Central University of Catalonia, 08500 Vic, Spain; elisenda.tarrats@uvic.cat; 2Department of Psychology, Faculty of Education, Translation, Sport and Psychology, University of Vic—Central University of Catalonia, 08500 Vic, Spain; 3Department of Social Psychology and Quantitative Psychology, Faculty of Psychology, University of Barcelona, 08007 Barcelona, Spain

**Keywords:** benevolent ethical climate, intrinsic motivation, affective commitment, burnout, depersonalization, Colombian electrical sector

## Abstract

Although affective commitment has been the focus of numerous studies, we know relatively little about certain factors that drive or hinder its progress. In this sense, this study contributes to the knowledge on the subject by establishing a relationship between a benevolent ethical climate and affective commitment, taking into account the mediating effect of intrinsic motivation. Furthermore, we highlight depersonalization as an aspect that can hinder these relationships when it assumes a moderating function. The sample was established through 448 employees of the Colombian electrical sector. The mediating effect was confirmed through a four-step method. The moderated mediation model was examined using SEM structural equations. The results show that a benevolent ethical climate is positively related to affective commitment and that intrinsic motivation is a mediating factor that justifies this relationship. However, depersonalization moderates the mediation between benevolent ethical climate, intrinsic motivation, and affective commitment. Specifically, the positive effect of the benevolent ethical climate on affective commitment is halted when depersonalization is high. The positive relationship between intrinsic motivation and affective commitment is interrupted when depersonalization is medium or high. Finally, as depersonalization progresses, the positive relationship between a benevolent ethical climate and intrinsic motivation is reduced. Therefore, organizations in the Colombian electrical sector must take measures that, in addition to avoiding social isolation, behave as indicators that warn when employees’ behaviors change significantly.

## 1. Introduction

Intrinsic motivation is a critical aspect for organizations to develop and continue. It is defined as the performance of a work activity without having to resort to external stimuli. That is, the task is sufficiently stimulating in itself to arouse the employee’s interest ([73], [74]).

[59] ([59]) establish a solid relationship between intrinsic motivation and affective commitment, as both concepts are directly linked to individual behavior. Specifically, affective commitment is the degree to which a person is emotionally committed to the organization and their main desire is to remain involved in its processes ([24]; [60]).

From this perspective, organizations and their managers need to enhance these positive aspects to ensure effective human resource management, and this favorable environment impacts both employee well-being and performance. One possible solution is to focus on contextual aspects such as the work climate. Indeed, there is growing interest in organizations promoting ethical work environments ([103]). The concept of an ethical climate is especially important as its main function is to transfer the necessary signals so that the employee understands the organization’s internal behavior ([55]; [77]).

In this sense, the impact of the ethical climate on intrinsic motivation and affective commitment has not been sufficiently analyzed ([11]; [18]; [38]; [43]; [83]). Therefore, the first objective of this study was to assess the relationship between ethical climate and affective commitment, taking into account the mediating effect of intrinsic motivation. To be more specific, we will focus on a particular ethical climate, which is benevolent, as several studies have highlighted its integrative character, making decisions based on the interest and protection of the entire community ([19]). Indeed, benevolence seeks options that maximize joint interests, even if this involves reducing the satisfaction of individual needs ([16]). In a broader sense, a benevolent ethical climate not only seeks the well-being of all employees within organizations but also cares about social interests in general. In reality, a benevolent ethical climate focuses more on results and the impact that organizational behavior has than on simple intentions that in the end do not have a significant repercussion on society ([10]).

[98] ([98]) establish that a benevolent climate is subject to a line of social reasoning as its main consideration is shared well-being. For this reason, it relies on friendship, group interest, and corporate social responsibility.

Friendship at work is an intentional relationship between two or more members of an organization that is sustained through emotional bonds. Its personal character contributes to trust being a critical factor that determines the intensity and attitude of the employee towards possible interactions with supervisors, colleagues, or subordinates. Consequently, friendship can intensify the employee’s emotional connection with the organization and with the tasks they develop. This approach increases the sense of belonging, motivation, and organizational commitment. Furthermore, friendship at work prevents unjustified absences, strengthens organizational support, and improves communication, aspects that directly influence intrinsic motivation and affective commitment ([35]; [41]; [54]).

On the other hand, cooperation and articulation between different work areas are essential characteristics within the work context. Therefore, group interest is a variable that has a significant specific weight in organizational progress ([6]). The concept of care and a work atmosphere based on common benefit builds a climate of trust that plays a prominent role for intrinsic motivation and affective commitment for advancement ([47]).

Corporate social responsibility also broadens organizational credibility and reputation. The interest in the environment and society as a whole are aspects that can motivate and engage the worker ([32]). The impact of corporate social responsibility shows the employee that the organization also takes their interests into account since it cares about providing fair conditions both internally and externally. This context can optimize intrinsic motivation and affective commitment ([1]; [22]; [51]).

However, even though a benevolent ethical climate may be positively related to affective commitment through intrinsic motivation, it is likely that people influenced by a benevolent climate are not always intrinsically motivated and consequently achieve optimal affective commitment. Therefore, it is important to analyze what circumstances or factors can moderate the mediating process between a benevolent climate, intrinsic motivation, and affective commitment.

During the last five years, the business landscape has been subject to profound changes, including the massive inclusion of decentralized work supported in virtual environments or the constant use of technologies. These changes in the way of working have forced the individual to a rapid behavioral adaptation to face new norms and requirements ([45]). Therefore, this disruptive situation can lead to feelings of social isolation that reach the extreme of depersonalizing the employee.

Depersonalization is a personal division experience characterized by severe disturbances in one’s sense of self. Depersonalization has a significant impact on the individual’s emotional and social quality of life. In fact, it is associated with parallel manifestations of anxiety, depression, or excessive burnout ([14]; [79]; [91]). For this reason, the second objective of this study was to assess whether depersonalization is a sufficiently important factor to hinder the possible relationship between a benevolent ethical climate and affective commitment mediated by intrinsic motivation.

This research is relevant from different points of view. First, there is a limited amount of research that has analyzed the relationship between ethical climates and affective commitment ([16]; [28]; [21]). Indeed, the possible association between a benevolent ethical climate and affective commitment has only been studied by [16] ([16]), [36] ([36]), [43] ([43]), and [70] ([70]). Although these authors have used samples that include public agency employees, higher education teachers, public accountants, and especially nurses, none included professionals linked with energy sectors, nor have any of these studies been conducted in a Latin American country. Moreover, most of these investigations have drawn a direct line between a benevolent ethical climate and affective commitment. Therefore, the underlying mechanisms involved in this relationship have gone unnoticed. Second, it is unknown when the link between a benevolent ethical climate and affective commitment increases and even in which situations that relationship stops. Therefore, intrinsic motivation may have a potential mediating effect in this association that the scientific community has not considered until now. Third, as far as we know, no study has examined the possible interrelationship between behavioral and emotional factors to explain the association between a benevolent ethical climate and affective commitment. In this sense, most of the research has not focused its efforts on studying the components of burnout separately, and depersonalization is the dimension that has the greatest effect on interpersonal relationships ([56]; [57]).

For all these reasons, the general objective of this study is to clarify the relationship between a benevolent ethical climate and affective commitment, introducing a model of moderated mediation, to improve the understanding of the underlying mechanisms involved in this relationship.

## 2. Study Contextualization

The Colombian electrical sector faces profound changes that go beyond the energy transition towards the use of cleaner energies. For example, some electric generation projects have delays of more than 10 years. For this reason, solving the lack of transparency in its management is the fundamental issue facing the Colombian electrical sector ([64]).

Moreover, the electrical sector has not been immune to the complications experienced by the Colombian economy during 2022 and 2023; geopolitical tensions and rising inflation put pressure on the country’s energy industry. In this sense, the change of government has been another decisive factor that has directly conditioned the external and internal policies of a sector such as the electrical one, which is predominantly publicly owned.

This complex and stressful environment has driven the electrical industry to strengthen the ethics of employees through good practices and climates that prioritize social interests. However, the new policies can affect not only the motivation and commitment of employees but also their emotional health ([81]). Therefore, there is a clear need in the electrical sector to understand whether an eminently social ethical climate, such as the benevolent one, positively impacts the intrinsic motivation and affective commitment of employees along with the underlying factors that may affect these possible relationships.

### 2.1. Benevolent Ethical Climate and Affective Commitment

The new regulations imposed by the Colombian government are putting strong pressure on its electrical sector. Specifically, there is a special interest in its companies operating responsibly. The Colombian electrical sector, since 2015, has been making progress through collective ethical action and transparency. This initiative seeks to rely on good practices not only to stop corruption in the sector but also to promote ethical environments based on strong corporate social responsibility (CSR). In this sense, a benevolent ethical climate promotes initiatives oriented towards people. Therefore, a work atmosphere of solidarity is likely to lead to a higher level of CSR ([10]; [76]).

When the employee perceives that the company they work for both prioritizes their well-being and cares about protecting the interests of others, both inside and outside the organization, they adjust their behavior to meet the ethical collective needs and commit to its correct application ([21]). In this sense, various studies have shown that a caring and supportive climate has a strong impact on organizational commitment ([27], [28]). In fact, fair institutional practices that promote a satisfying work experience have an impact on the employee’s ability to develop an emotional bond with the sector to which they belong ([34]). Therefore, a benevolent ethical climate that establishes codes of conduct directed towards complicity, collective trust, and interest in social needs increases the employee’s feelings of identification with the organization. That is, it expands their affective commitment and the desire to remain connected with the different business processes ([18]).

The self-determination theory (SDT) proposed by [17] ([17]) and [72] ([72]) starts from the hypothesis that human beings are implicitly predisposed to grow and emotionally integrate with others. However, for this to happen, specific supportive conditions are needed. Self-determination theory clearly distinguishes between autonomous forms of motivation and controlled forms of motivation. Autonomous forms of motivation consist of intrinsic motivation, integrated motivation, and identified motivation, whereas controlled forms of motivation include introjected regulation and external regulation. People in a positive social context tend to experience autonomous motivation through a sense of personal choice, interest, and volition, which helps them to assimilate or internalize their personal values and increase the quality of their motivation ([17]). One of the social contexts that can influence employee autonomous motivation is a benevolent work climate. Therefore, a work climate that cares for the employee becomes an inducing force that in addition to increasing autonomous motivation also directly influences the employee’s affective commitment. In fact, motivation and commitment have a reciprocal relationship and are positively associated.

In this sense, social exchange theory (SET) represents another theoretical assumption that studies how social relationships are formed ([9]). It is based on the idea that relationships are built on the basis of comparison with alternatives and the use of a cost–benefit analysis. Consequently, a benevolent ethical climate that is concerned with personal feelings is conducive to affective employee commitment through reciprocal relational responses ([69]). Therefore, the following hypothesis is proposed:

**H1.** 
*A benevolent ethical climate is positively related to affective commitment.*


### 2.2. Mediating Effect of Intrinsic Motivation

SDT maintains that adequate personal development depends on the constant support of the three central psychological needs: autonomy, competence, and relationship. This theory was initially entirely oriented towards intrinsic motivation as an example of the inclusive tendencies of human nature ([73], [74]).

In this sense, a benevolent ethical climate can be that contextual factor that drives those three primary needs through its different dimensions, especially friendship, group interest, and CSR. In fact, a favorable social environment significantly affects the employee’s subjective perception of the intrinsic importance of work ([90]). For this reason, the friendships that are born in the work circle, being voluntary and personal, are more than likely to be a source of social and emotional resources that influence the employee’s motivation. Interpersonal support, the possibility of openness, and the balance offered by friendship are transformed into joint efforts usually directed towards the mutual achievement of objectives ([54]). Therefore, friendship induces not only greater self-esteem but also a strong perception of autonomy, competence, and relationship, which are determining aspects in intrinsic motivation ([25]).

On the other hand, group interest develops through constant interpersonal interactions that generate closeness and relationships. Indeed, various studies have shown that affinity with others affects intrinsic motivation, in the sense that individuals strive in their relationships when they feel them to be authentic, which also increases their sense of belonging ([62]).

Likewise, the Colombian electrical sector faces significant challenges that not only affect environmental responsibility, but also equality of opportunities in labor insertion. CSR initiatives in Colombia are betting on decarbonization and an energy transition in which the electrical sector is taking a leading role. In fact, there are still about 400,000 households in the country without electricity. Therefore, CSR, by showing an interest in social problems, represents an opportunity for employees to increase their intrinsic motivation through easily understandable ethics. That is, one that relies on what is universally considered correct ([3]). In addition, the positive relationships between an organization and its environment suggest to the employee that their work is carried out in a safe environment. In this sense, security is an important precursor for intrinsic motivation to progress by satisfying the need for relationship ([51]).

Both intrinsic motivation and affective commitment are resources that have strong consequences on behavior. Affective commitment requires the employee to identify clearly with the organizational objectives and values. These factors are also necessary for intrinsic motivation to progress ([31]). Indeed, the positive results of workers with strong intrinsic motivation include an advancement of affective commitment ([30]). Therefore, motivation is a primary component through which commitment is promoted. That is, interest in the task strengthens the employee’s emotional bond with the organization ([4]). Finally, SET provides a basic theoretical basis for describing the nature of the relationship between the employee and the organization. An essential element of SET is its ability to engender trust, which directly influences employee intrinsic motivation ([2]). This is because the high-quality relationship associated with trust allows employees to focus on their jobs and assume the responsibility that comes with it, which generates high intrinsic motivation. Based on the above reasoning, we proposed the following hypothesis:

**H2.** 
*Intrinsic motivation mediates the positive relationship between a benevolent ethical climate and affective commitment.*


### 2.3. Moderating Effect of Depersonalization

The extreme changes in the use of technology, adopted by the Colombian electrical sector since 2020, force its organizations to remain in permanent change management, which directly affects the employee. Indeed, decentralized work with low interdependence has become an alternative to increase organizational competitiveness ([45]). In this sense, social support and interaction, which are critical aspects for emotional well-being, may blur to the point where the individual isolates themselves and experiences a strong sense of indifference to the environment.

SET reveals that reciprocity plays a central role in human life. Therefore, people seek fairness in their social relationships since the imbalance between effort and reward can lead to a spiral of loss of resources that disconnects the individual emotionally. In fact, depersonalization usually occurs when the effort invested is inversely proportional to the reward received ([85]).

On the other hand, the theory of human motivation within SDT provides an ideal framework for understanding how the satisfaction of the basic psychological needs of autonomy, competence, and relatedness influence people’s cognitive well-being. In that sense, when the individual is not able to cope effectively with stress due to a lack of psychological resources, work challenges become threats that wear the employee down emotionally to the point of separating him from his own experiences. This context of constant demands often depersonalizes the worker ([89]).

Depersonalization causes the employee to gradually distance themselves from work-related issues, avoid their responsibility, and distance themselves from their colleagues to such an extent that their attitude ignores the needs of others and stops the motivation for the activity they perform ([67]). In addition, it is associated with other symptoms such as anxiety, psychosomatic disorders, chronic fatigue, or depression. Depersonalization is not just a continuous state in which the individual drastically alters the way they respond to certain environmental demands or obstacles ([15]). Therefore, it is more than likely that a person who blocks certain emotions avoids commitment and disregards the intrinsic value of the task.

For example, [8] ([8]) consider that intrinsic motivation favors adaptive behavior. Therefore, the purpose of the task and its relevance protect the individual from possible depersonalization. However, the incompatibility between personal needs and the work environment is likely to result in a drastic reduction in intrinsic motivation ([71]). Indeed, the characteristics of a benevolent ethical climate can hardly influence intrinsic motivation or affective commitment when depersonalization moderates these relationships. Specifically, when depersonalization reaches a medium or high level, the employee may experience a strong deterioration in the perception they have of friendship at work and also reduce their interest in group needs or the social responsibility of the organization ([63]).

In addition, depersonalization can lead to emotional collapse ([95]). This scenario of constant crisis generates rigid and apathetic behaviors towards people’s feelings along with certain contempt for any of their needs. At this point, interpersonal relationships become a demand that the individual is unable to assume. Therefore, depersonalization can break any type of emotional bond and also decrease aspects closely linked to intrinsic motivation and affective commitment such as level of vital energy, degree of concentration, ability to persist in a task, ability to face failure, and self-esteem itself ([65]).

The conservation of resources (COR) theory suggests that people invest a large amount of emotional energy to prevent the active loss of resources. This coping strategy forces the employee to use additional resources to prevent chronic stress ([39]). Therefore, highly depersonalized people are unlikely to possess enough emotional capacity for intrinsic motivation to develop and also influence affective commitment. The results of [88] ([88]) support our argument by concluding that depersonalization is an obstacle to the development of autonomous motivation.

On the other hand, affective commitment is an important resource to be able to cope with the negative effects of depersonalization since it is strongly associated with a perception of social support ([87]). Indeed, the theory of social identification indicates that affectively committed people perceive the organization as part of their own identity. However, affective commitment needs a constant replenishment of emotional and social resources. For this reason, when depersonalization progresses, work performance declines, as the individual stops supplying valuable resources, and in return, demands disproportionately increase ([53]). This negative context influences unjustified absenteeism and has alarming effects on levels of organizational commitment ([92]).

Based on these arguments, we propose the following hypotheses:

**H3a.** 
*Depersonalization moderates the positive relationship between a benevolent ethical climate and intrinsic motivation, so that when depersonalization is high, the positive relationship between a benevolent ethical climate and intrinsic motivation is significantly reduced.*


**H3b.** 
*Depersonalization moderates the positive relationship between intrinsic motivation and affective commitment, so that when depersonalization is high, the positive relationship between intrinsic motivation and affective commitment stops.*


**H3c.** 
*Depersonalization moderates the positive relationship between a benevolent ethical climate and affective commitment, so that when depersonalization is high, the positive relationship between a benevolent ethical climate and affective commitment stops.*


## 3. Presentation of the Study

This research reviewed the relationship between a benevolent ethical climate and the affective commitment of employees of the Colombian Electrical Sector. Secondly, this research verified whether intrinsic motivation acted as a mediating construct of the relationship between a benevolent ethical climate and affective commitment. Finally, this study verified whether depersonalization could moderate the effect of a benevolent ethical climate on intrinsic motivation and affective commitment, along with the independent effect of intrinsic motivation on affective commitment. Figure 1 represents the moderated mediation model.

### 3.1. Methods

#### 3.1.1. Participants

The study included 448 professionals belonging to six organizations of the Colombian Electrical Sector. Following the suggestions proposed by [46] ([46]) or [86] ([86]), a probabilistic cluster sampling method was used with a 95% confidence level. This technique is particularly useful as specific departments (clusters) can be studied. The cities with the largest concentration of companies in the analyzed sector were purposefully examined. The effective response rate of the questionnaire was 100% since the participating organizations designated specific times within working hours to accommodate the employee and ensure valid responses. Overall, 61% of the total sample were men (273). The average age was 37.18 (SD = 10.06), with a range of 20 to 69 years. The average number years in work was 13.06 (SD = 8.82), ranging from 1 to 38. As for education, 100% held university studies, and 57.42% held master’s degrees or doctorates. Regarding the type of contract, 100% had indefinite contracts. Finally, 42.40% of the participants had no children.

#### 3.1.2. Measures

##### Benevolent Ethical Climate

To measure the benevolent ethical climate, the 11-question scale suggested by [98] ([98]) was used. It is divided into three subscales: (1) Friendship (3 items); (2) Group interest (4 items); and (3) Social responsibility (4 items). Recent studies have confirmed its reliability and validity ([100]). Some of the items used were “The most important concern is the well-being of all people in the company”. Each item was measured from 1 (totally disagree) to 6 (totally agree). The present research obtained a Cronbach’s alpha of 0.88. Likewise, composite reliability (CR) and average variance extracted (AVE) were calculated. The results indicate that CR is optimal (CR = 0.74), and AVE is adequate (AVE = 54%). According to [5] ([5]), these two values are relevant as they are above 0.70 and 50%, respectively.

##### Intrinsic Motivation

To measure intrinsic motivation, the 5-question scale suggested by [96] ([96]) was used. Recent studies have confirmed its reliability and validity ([23]). Some of the items used were “I enjoy finding solutions to complex problems”. Each item was measured from 1 (totally disagree) to 6 (totally agree). This research reaches a Cronbach’s alpha of 0.90. Additionally, the results indicate that CR is optimal (CR = 0.73), and AVE is adequate (AVE = 53%).

##### Affective Commitment

To measure affective commitment, the 6-question scale suggested by [58] ([58]) was used. Recent studies have confirmed its reliability and validity ([66]). Some of the items used were “I really feel the problems of the organization as my own”. Each item was measured from 1 (totally disagree) to 6 (totally agree). This research reaches a Cronbach’s alpha of 0.87. Additionally, results indicate that CR is optimal (CR = 0.76), and AVE is adequate (AVE = 79%).

##### Depersonalization

To measure depersonalization, the 5-question scale suggested by [85] ([85]) was used. Recent studies have confirmed its reliability and validity ([84]). Some of the items used were “I just want to do my job and not be bothered”. Each item was measured from 1 (totally disagree) to 6 (totally agree). Following the recommendations of [75] ([75]), item number thirteen of the scale was removed. The present research achieved a Cronbach’s alpha of 0.90. The results indicate that CR is optimal (CR = 0.86), and AVE is adequate (AVE = 66%).

#### 3.1.3. Procedure

The link with the Colombian Electrical Sector was established through the collective action of ethics and transparency promoted by the XM company in 2021. In reality, through a presentation and different suggestions, the objectives of this research were established. The validation of the questionnaire along with the scales used was carried out through a group of specialists proposed by the organizations studied. The research project was approved by an ethics committee in July 2021. The intervening organizations were sent, by email, a series of documents to guarantee confidentiality, authorization, data protection, and possible voluntary abandonment of each of the participants. The survey processes were carried out on independent days ([68]) and each organization designated around sixty minutes to guarantee a significant number of responses.

#### 3.1.4. Data Analysis

First, potential outlier data that may affect the analysis of the results are identified through the probability identifier (<0.01) with the statistical program SPSS v.25. To determine the normality of the variables, the values of skewness and kurtosis are defined, the different variables are below 2, which, according to [49] ([49]), shows normality. Second, through the test of homogeneity of variances, it is deduced that there is homoscedasticity since (*p* > 0.05). Third, multiple regression analyses are performed using the PROCESS v. 3.5 macro. The structural equation model number 59 proposed by [37] ([37]) (moderated mediation) with a 95% CI and bootstrapping sampling of 10,000 through the AMOS v. 26 macro is used. Fourth, to review the collinearity problems, it is verified that the VIF indices are less than 5.

## 4. Results

### 4.1. Measurement Model Evaluation

Given that the study is proposed through a model with several latent constructs, it is pertinent to evaluate its reliability (internal consistency) and validity (convergence and discrimination) ([50]), followed by the execution of a structural equation model SEM to test the study results. The CFA, which is the confirmatory factor analysis (measurement), and the SEM (structural) were drawn and executed in AMOS v.26. The CFA is carried out using the following absolute fit indicators: (χ^2^), likelihood ratio; (χ^2^/gl), chi-square with respect to degrees of freedom; GFI, goodness of fit index; RMSR, root mean square residual; RMSA, root mean square error of approximation. These values indicate the degree to which the model can anticipate the covariance matrix analyzed. At the same time, incremental adaptation factors are used: IFI, incremental fit index; NFI, normed fit index; CFI, comparative fit index. These values differentiate the suggested model from another that generally does not explain the relationship between the variables. The CFA confirms the validity of the theoretical model proposed. Results: χ^2^ = 711.35, *p* < 0.01; χ^2^/gL = 2.34; GFI = 0.921; IFI = 0.948; NFI = 0.938; CFI = 0.947; RMSEA = 0.0541; RMSR = 0.0538. It is therefore concluded that the adjustment is acceptable. χ^2^/gL < 3; IFI, NFI, GFI, CFI > 0.90; RMSEA < 0.006; RMSR < 0.008 ([42]).

### 4.2. Descriptive Statistics and Discriminant Validity

The means, standard deviations, and correlations between the research variables are presented in Table 1. The benevolent climate showed negative correlations with depersonalization (r = −0.19, *p* < 0.01) and positive with intrinsic motivation (r = 0.36, *p* < 0.01), and affective commitment (r = 0.34, *p* < 0.01). Since the benevolent climate was positively correlated with affective commitment, Hypothesis 1 was accepted. Depersonalization showed negative correlations with intrinsic motivation (r = −0.21, *p* < 0.01) and with affective commitment (r = −0.57 *p* < 0.01). Intrinsic motivation showed significant positive correlations with affective commitment (r = 0.29, *p* < 0.01). *N*, number of items; M, mean; SD, standard deviation.

To determine discriminant validity, the following actions were taken. First, the values of the average extracted variance (AVE) were determined, which represent the total variance of the indicators captured by the latent variable. All variables showed AVE values close to 0.5 ([5]). Second, the square root of AVE was determined. According to [26] ([26]), a latent construct has sufficient divergence from other latent constructs when the value of the square root (values in bold on the diagonal of Table 1) is greater than its bivariate correlation coefficient with other latent constructs. Based on the different results, it was concluded that there is solid discriminant validity between the analyzed constructs.

### 4.3. Mediating Effect Analysis

In Hypothesis 2, we anticipated that intrinsic motivation would mediate the relationship between a benevolent ethical climate and affective commitment. To examine this hypothesis, we followed a four-step procedure to establish the mediating effect ([7]) which requires the following: (a) a significant association between the benevolent ethical climate and affective commitment; (b) a significant association between intrinsic motivation and the benevolent ethical climate; (c) a significant association between intrinsic motivation and affective commitment after controlling for the benevolent ethical climate; (d) a significant coefficient for the indirect path between the benevolent ethical climate and affective commitment through intrinsic motivation. The percentile bootstrapping method with bias correction determines whether the last condition is met. The regression analysis revealed that, in the first step, the benevolent ethical climate positively predicted affective commitment, β = 0.19, *p* < 0.01 [0.18; 0.64] (see Model 1, Table 2). In the second step, the benevolent ethical climate positively predicted intrinsic motivation, β = 0.24, *p* < 0.01 [0.07; 0.38] (Model 2, Table 2). In the third step, after controlling for the benevolent ethical climate, intrinsic motivation positively predicted affective commitment, β = 0.31, *p* < 0.01 [0.14; 0.54] (Model 3, from Table 2). Lastly, the percentile bootstrapping method corrected for bias showed that the indirect effect of the benevolent ethical climate on affective commitment through intrinsic motivation was significant (ab = 0.24 × 0.31 = 0.07, SE = 0.02, 95% CI = [0.11; 0.44]). The mediating effect accounted for 32.67% of the total effect. In general, the four previous criteria were met to establish a mediating effect, supporting Hypothesis 2.

### 4.4. Moderated Mediation Analysis

In Hypothesis 3, this study assumed that depersonalization would moderate the association between the benevolent ethical climate and affective commitment, the benevolent ethical climate and intrinsic motivation, and intrinsic motivation and affective commitment. The moderating effects of depersonalization on the relationship between the benevolent ethical climate and intrinsic motivation (Model 1), the relationship between the benevolent ethical climate and affective commitment (Model 2), and the relationship between intrinsic motivation and affective commitment (Model 3) were estimated. The specifications of the three models are summarized in Table 3 and Figure 2.

The results revealed that the relationship between a benevolent ethical climate and intrinsic motivation was moderated by depersonalization (b = −0.01, SE = 0.02, 95% CI = [−0.12, −0.36]) (Table 3, Figure 2). In addition, Figure 3 explains the detail of the moderation. As the individual feels greater depersonalization, the impact of the benevolent ethical climate on intrinsic motivation decreases. The high, medium, and low effects of depersonalization are significant. Therefore, H3a is accepted.

In second place, the relationship between intrinsic motivation and affective commitment was moderated by depersonalization (b = −0.02, SE = 0.01, 95% CI = [−0.09, −0.54]) (Table 3, Figure 2). In addition, Figure 4 explains the detail of the moderation. At medium and high levels of depersonalization, the association between intrinsic motivation and affective commitment is interrupted. Only low depersonalization allows intrinsic motivation to influence affective commitment. The medium and high effects of depersonalization are not significant. Therefore, H3b is accepted.

In third place, the relationship between a benevolent ethical climate and affective commitment was moderated by depersonalization (b = −0.01, SE = 0.03, 95% CI = [−0.05, −0.36]) (Table 3, Figure 2). In addition, Figure 5 explains the detail of the moderation. At low levels of depersonalization, the benevolent ethical climate influences affective commitment. At medium levels of depersonalization, the effect between both variables is almost irrelevant. At high levels of depersonalization, the benevolent ethical climate ceases to have an impact on affective commitment. Therefore, H3c is accepted.

## 5. Discussion

This study explored the relationship and the underlying mechanisms that increase or decrease the distance between the benevolent ethical climate and affective commitment in Colombia. The results showed that the benevolent ethical climate was positively related to the affective commitment of employees of the Colombian electrical sector. Intrinsic motivation played a mediating role between a benevolent ethical climate and affective commitment. Lastly, depersonalization moderated the mediating effect. These findings are discussed in detail below.

The first significant result showed that a benevolent ethical climate is positively related to affective commitment (H1), which is consistent with previous studies, e.g., [16] ([16]) and [70] ([70]). On the other hand, [28] ([28]) and [97] ([97]) consider that a caring ethical climate positively influences organizational commitment. However, the former authors use a different measurement scale, specifically that of [20] ([20]), and the latter focus on a climate of concern for others but do not take into account the specific dimensions of friendship, group interest, and corporate social responsibility (CSR).

A benevolent ethical climate, in essence, seeks alternatives that expand common interests and decrease individual needs. For this reason, friendship at work is a factor that encourages the employee to intentionally assume shared obligations along with other colleagues. Personal development through strong emotional bonds improves the feeling of belonging and promotes affective commitment ([41]). Indeed, the social support that comes from friendship builds a positive work environment that not only impacts affective commitment but also reduces employee turnover and improves performance ([35]). The emotional identification with the group facilitates common values and objectives that are an essential part of affective commitment ([82]). On the other hand, a benevolent ethical climate deliberately seeks the stability of the members of the organization. This social process, of interest in others, establishes a series of affective signals and goodwill that become indicators of greater affective commitment.

Cohesion is the result of personal effort associated with a supportive climate. For this reason, similar groups commit to organizational objectives and are more oriented towards cooperation and positive feelings ([16]; [55]; [98]). Finally, a socially responsible organizational system influences individual trust levels and personal attitudes. Therefore, it is reasonable to think that a work climate that pursues well-being, cooperation, and solidarity, both internal and external, captures the employee’s emotional interest ([1]). Social exchange theory ([48]) is a theoretical framework that explains why CSR and affective commitment are connected. In this sense, exchange relationships involve investments that engage all stakeholders. That is, CSR is a tool that, due to its social orientation, captures the employee’s attention and their response is greater organizational identification by feeling co-participant in numerous beneficial actions for the community in general ([12]).

The second significant result of this research revealed that intrinsic motivation mediates the positive relationship between a benevolent ethical climate and affective commitment (H2). As far as we know, this is the first study to establish this type of association.

Friendship at work increases the perceived value of tasks. That is, work friendship goes beyond satisfying certain emotional needs and impacts the employee’s attitude by raising their intrinsic motivation. Friendship relationships are characterized by exchanging resources and by a common openness that brings with it greater proactivity and interest in the job ([54]). Indeed, friendship is a positive experience that has immediate benefits, and according to Fishbach and Woolley, this inner sensation expands intrinsic motivation and the ability to persist in achieving a goal.

On the other hand, intrinsic motivation comes from different sources. The interest in other people involves a prosocial approach that aims to increase their well-being. This closeness and concern for the group make the employee enjoy the process (intrinsic motivation) and at the same time consider that the result has a positive impact on others ([29]). Therefore, the interest in the group and intrinsic motivation are associated through the perception of autonomy, competence, and relationship ([78]). In fact, recent research from the theory of self-determination (SDT) confirms that prosocial motivation tends to satisfy three basic psychological needs ([17]; [74]; [101]). Specifically, the employee perceives autonomy by acting voluntarily for the benefit of others, competence by providing effective support to others, and relationship by being able to connect actions with results that have a positive impact on the lives of certain people ([33]).

Finally, the commitment to CSR has a positive impact on intrinsic motivation. Indeed, working in a socially responsible organization allows the employee to share the values and social awareness of the company ([44]). In addition, the intersection of principles builds a work environment where priorities are usually autonomy and empowerment, which are resources that directly influence intrinsic motivation. In conclusion, actively pursuing social change affects the employee’s level of contribution and proactivity. Likewise, intrinsic motivation actively promotes affective commitment. Intrinsically motivated employees develop feelings of identification and belonging that are the foundations from which affective commitment progresses ([31]). Indeed, both intrinsic motivation and affective commitment are complementary factors as they drive participation, autonomy, commitment to objectives, and the achievement of organizational results ([45]).

The third result of this research found that depersonalization played a moderating role in the mediating mechanism (H3a, H3b, and H3c). Specifically, depersonalization impairs the positive effects of the benevolent ethical climate on intrinsic motivation (H3a). Depersonalization impairs the positive effects of intrinsic motivation on affective commitment (H3b). Depersonalization impairs the positive effects of a benevolent ethical climate on affective commitment (H3c). These results supported the argument that depersonalization is a risk factor for organizations in an era marked by digitization, excessive work climates, and excessive emphasis on organizational objectives rather than the care of interpersonal relationships. Ethics and the context it builds, intrinsic motivation, and affective commitment are variables that are related to the emotional regulation of the individual and consequently are positively associated with job satisfaction, performance, low turnover, and personal fulfillment. However, when depersonalization acts as a moderating factor, the individual attenuates his or her subjective emotional experiences that span the entire affective and contextual range. Therefore, depersonalization prevents the employee from connecting with his emotions, which prevents him from being able to commit himself to the organization and also from feeling that his work activity is a focus of learning or an incentive for his own personal growth. In fact, depersonalization remains a poorly understood and poorly contextualized phenomenon.

Firstly (H3a), we found that depersonalization moderated the relationship between the benevolent ethical climate and intrinsic motivation. This association weakened as the level of depersonalization increased. This result suggests that depersonalization prevents the employee from feeling that friendships or personal needs are something relevant or satisfying. Depersonalization implies that the individual shows apathy and disinterest in the feelings of other people. Indeed, depersonalization is associated with emotional rigidity, low self-esteem, and low self-confidence ([102]). Therefore, the individual avoids relating and ultimately perceives the rest of the employees as objects. This emotional paralysis prevents depersonalization from being managed through intrinsic motivation since the person loses self-efficacy and interest in their role in their organization ([61]).

Secondly (H3b), we evidenced that depersonalization had a moderating effect on the positive relationship between intrinsic motivation and affective commitment. This association ceased to be significant at medium and high levels of depersonalization. This result suggested that depersonalization could drastically reduce the use of certain personal resources, for example, intrinsic motivation. Intrinsic motivation is a driving force that motivates the employee to commit to their work functions and organizational objectives. However, according to the theory of COR ([39]; [40]) individuals who lose the ability to self-regulate emotionally consume a large amount of resources until they reach depersonalization. In that situation, the positive effect of intrinsic motivation on affective commitment is reduced until it stops ([52]).

Thirdly (H3c), depersonalization moderated the association between the benevolent ethical climate and affective commitment. The relationship became non-significant when the individual experienced high depersonalization. This result suggests that depersonalization prevents the employee from perceiving that personal interaction or interest in society are positive aspects that guide emotions towards the organization and its objectives. The employee with strong depersonalization not only experiences tiredness, low mood, frustration, or a perception of emotional overload but also distances themself from others ([56]). In that sense, negative emotions are associated with dysfunctional behaviors that cause changes in the emotional load capacity (ECC) and in turn in affective commitment. Although the ability to openly express emotions, positive or negative, can improve affective commitment ([13]), the own inability, associated with depersonalization, to openly express the emotional state usually leads to a greater escalation of disappointment and a feeling of failure that deteriorates affective commitment ([93]; [95]).

### 5.1. Practical Implications

In general, our study suggests that organizations in the Colombian electrical sector need to continue implementing policies that support their employees from a perspective that goes beyond superficiality and individual interests. The executive managers who lead the sector need to maintain an ethical culture that strengthens the idea that these practices are sincere and not merely designed to establish processes with little reach. Indeed, the social exchange that comes from the perception of integrity captured by the employee leads to higher levels of intrinsic motivation and affective commitment.

Firstly, one way for the employee to feel that the organization cares about their well-being is to adapt internal policies that help family reconciliation, for example, extending maternity or paternity leave ([103]). Secondly, promoting social interaction in highly digitized work environments prevents isolation or high perception of stress that usually translate into lower levels of intrinsic motivation or affective commitment. Indeed, friendship at work is a social resource that directly influences employee turnover or unjustified absenteeism ([41]). Likewise, the possibility of including support groups prevents excessive work pressure and builds an open avenue for the employee to communicate over short periods of time. Therefore, organizational support can be transformed into a useful psychological resource for employees of the Colombian electrical sector that also reduces a possible emotional disconnection ([104]).

Thirdly, limiting competition within organizations will result in greater intrinsic interest in the task and surely in growing affective commitment. Understanding not only the capabilities of the employees but also their aspirations will accelerate internal selection processes. When the employee perceives that the organization values effort and competencies from a personal and objective approach, it increases the feeling of equality, which prevents competition from harming relationships within the organization ([54]). Fourthly, corporate social responsibility (CSR) initiatives go beyond laws and social norms. The Colombian electrical sector has the opportunity to use CSR as an ongoing strategy that observes and evaluates the relationship between employee and employer to achieve a greater impact on intrinsic motivation and affective commitment. Specifically, recognizing the origin of the worker and their socioeconomic situation will allow the sector to implement internal CSR initiatives that assign support resources for medical treatments or specific personal situations ([3]).

Fifthly, the most important finding of this research is that depersonalization compromises the mediating mechanism that relates a benevolent ethical climate with intrinsic motivation and affective commitment. Therefore, depersonalization is a key factor that influences the effect of a benevolent ethical climate. The ability to monitor the emotional state of employees is crucial for the healthy development of organizations. In that sense, the Colombian electrical sector can hold workshops where the employee can develop emotional regulation patterns. Specifically, strategies include those focused on antecedents and not so much on the response. The former influence an early stage of emotional evolution and modify the emotion before it occurs in its entirety. The latter seeks concrete action plans once the emotion has already been triggered. That is, an adjustment phase is initiated between the context and the feeling ([80]).

Very recent studies conclude that in clinical settings the diagnosis of depersonalization is severely neglected since experts tend to confuse depression and depersonalization due to the large number of similar characteristics between both disorders. Therefore, organizations can implement targeted interventions to primarily diagnose depersonalization, e.g., through structured interviews for dissociative disorders ([102]). Some common symptoms of depersonalization are related to the inability to link memories and emotions, that is, with constant emotional insensitivity or with the loss of control over movement and the way of expressing oneself ([99]).

Lastly, one of the tragic paradoxes of depersonalization is that the most susceptible people seem to be the most dedicated, conscientious, responsible, and motivated. People with these traits tend to be idealistic and perfectionistic, which can lead them to immerse themselves in their work and devote themselves to it to the point of being unable to contribute anything else. Therefore, the Colombian electricity sector needs to develop workshops that help employees in the proper use of autonomy in order to be able to adequately reconcile personal and professional life. In fact, long working hours and the lack of control over them become counterproductive habits to achieve a full and balanced life.

In addition, the Colombian electricity sector and specifically its managers need to implement mentoring strategies that promote a balanced lifestyle. Promoting personal growth and renewal implies optimizing the sense of meaning at work and in personal life. In this sense, the Colombian electricity sector can include strategies that contribute to employee well-being through renewing and energizing activities, such as continuous training activities outside of work or informal meetings with other members of the organization ([83]).

### 5.2. Limitations and Future Research

The present study was subject to some limitations. First, data collection was carried out through a self-report survey that may involve possible biases, e.g., social desirability or the difficulty of being able to generalize the results obtained. However, the sample being probabilistic by clusters is a reason for greater representativeness. Moreover, this study used a cross-sectional design to analyze the relationship between a benevolent ethical climate and affective commitment. Due to the cross-sectional nature of the data, there may be doubts regarding causal inferences. Therefore, longitudinal research is needed to examine, more accurately, the causal relationship between the variables. Likewise, the sample for this study was predominantly male due to the characteristics of the job. Therefore, research with more balanced samples in terms of gender is required to draw solid conclusions. Finally, the common method variance (CMV) could be attenuated since data were collected through six different sources ([68]).

Future research can explore the effect of emotional exhaustion, which is a common reaction among employees who are subject to strong work pressures. Although a benevolent ethical climate aims to reduce burnout through social exchange and shared learning, more research is still needed to confirm this buffering role. On the one hand, there are indications that affective commitment and well-being are intertwined, but it is still unknown if emotional exhaustion can be an underlying mechanism that interrupts that relationship ([94]). On the other hand, this research advanced the knowledge of a benevolent ethical climate and its link to affective commitment using mediating and moderating mechanisms. However, there are significant variables at the personal or organizational level, e.g., emotional intelligence, self-efficacy, and psychological empowerment, that seem to be related to affective commitment and could moderate that association. In addition, including covariates such as gender or job tenure may result in a broader perspective of the attenuating or driving effect of these two factors.

Future research can mitigate the biases inherent in cross-sectional design by avoiding selection bias or recall bias. Selection bias is an error that occurs when choosing individuals or groups to participate in a study. This can lead to the results of the study not accurately representing the population to be analyzed. Selection bias can be prevented in the design phase of the study. To do this, care should be taken to ensure that the subjects in a study are similar to each other and to the general population. In addition, the sample size should be representative. On the other hand, recall bias is a systematic error that occurs when people do not accurately recall a past event or experience. To prevent recall bias, techniques such as pilot tests, interviews, and focus groups can be used.

In addition, bias associated with a self-report survey can be prevented by avoiding biased, leading, compound, ambiguous, or absolute questions. Multiple-choice questions provide a more controlled approach to collecting perceptions as long as the responses are inconclusive. Future research can take all of these suggestions into account.

Finally, future research could explore whether there are differences in behaviors between workers in private companies and those in public entities. It is likely that work pressure varies in private and public contexts, eliciting different responses that reduce or enhance the impact of ethical climates. In addition, the long-term effects of depersonalization have almost no contrasting studies. Future research may review the extent to which depersonalization impairs employees’ intrinsic motivation and affective commitment over long periods of time. In fact, a selfish ethical climate, a principled ethical climate, and a benevolent ethical climate propose different strategies and messages. Therefore, their relationship with depersonalization may be totally different.

## 6. Conclusions

Taking into account the general objective of this study, which was to clarify the relationship between a benevolent ethical climate and affective commitment through a model mediated by intrinsic motivation and moderated by depersonalization, we arrived at the following conclusions.

A benevolent ethical climate plays a key role in organizations oriented towards social responsibility. Taking into account the potential effect of an action on others enables an analysis process that, in addition to influencing decision making, strengthens interpersonal ties. The results suggest that the Colombian electrical sector has been able to build a conciliatory climate in which personal values and community well-being are priority aspects. A benevolent ethical climate insists on developing positive relationships between employer and employee that reinforce trust and reduce uncertainty. When the employee believes that their employer possesses integrity and ethical values, the necessary conditions are established to promote intrinsic motivation and affective commitment. The individual conviction that the organization will fulfill its promises and continue to protect the employee’s job, regardless of the situation, builds strong emotional ties that directly impact intrinsic motivation and affective commitment.

However, the positive effect of a benevolent ethical climate on intrinsic motivation and affective commitment depends on the degree of depersonalization. Specifically, the positive influence of the benevolent ethical climate on intrinsic motivation weakened, and the positive effect on affective commitment, disappeared when depersonalization was high. In addition, depersonalization, when it reached medium or high values, stopped the positive relationship between intrinsic motivation and affective commitment. These findings suggest that emotional factors, resulting from severe stress, have the capacity to interrupt or limit essential aspects in organizational development. The constant feeling of detachment triggers feelings of low personal acceptance that are incompatible with intrinsic motivation and affective commitment.

## Figures and Tables

**Figure 1 ejihpe-15-00055-f001:**
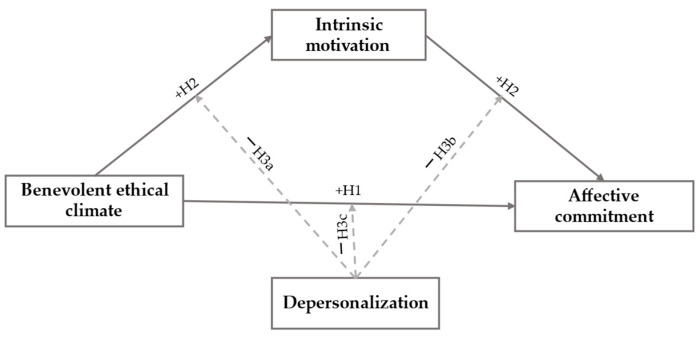
The proposed moderated mediation model.

**Figure 2 ejihpe-15-00055-f002:**
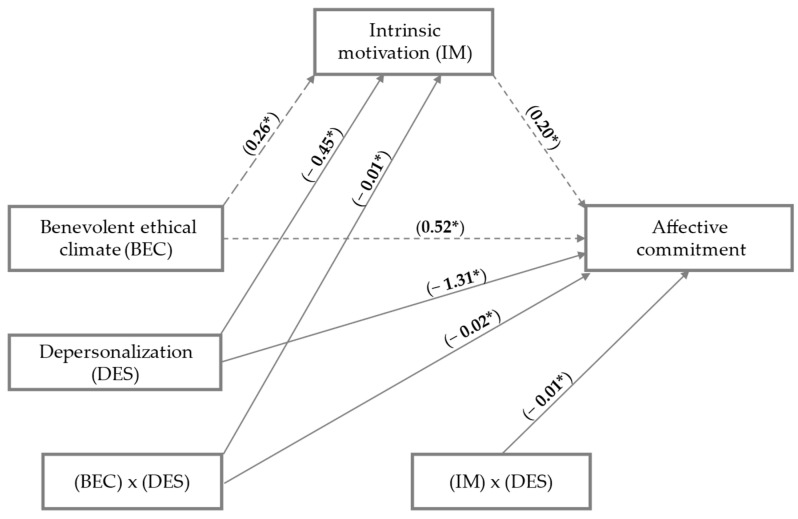
The proposed moderated mediation model unstandardized coefficients. * *p* < 0.05.

**Figure 3 ejihpe-15-00055-f003:**
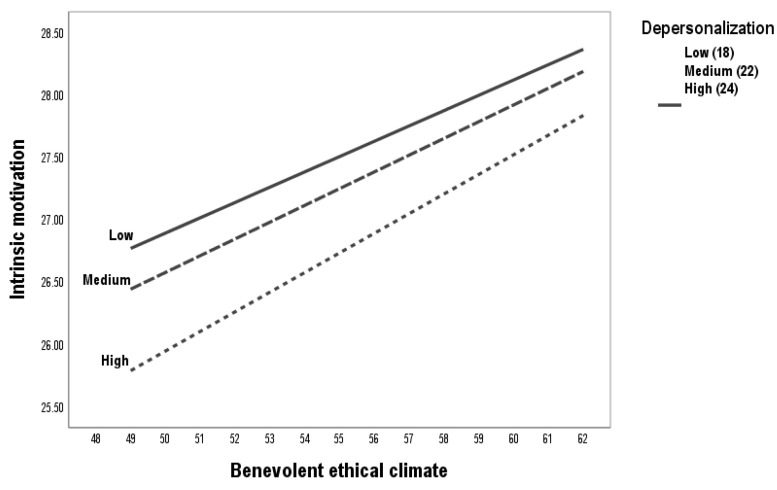
Moderation process between benevolent ethical climate and intrinsic motivation.

**Figure 4 ejihpe-15-00055-f004:**
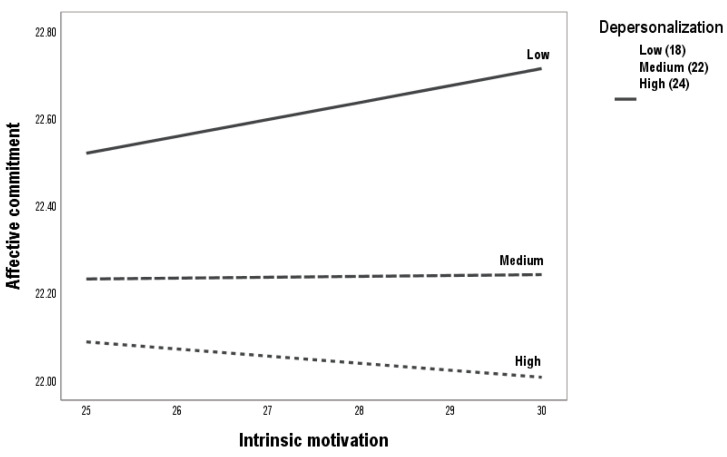
Moderation process between intrinsic motivation and affective commitment.

**Figure 5 ejihpe-15-00055-f005:**
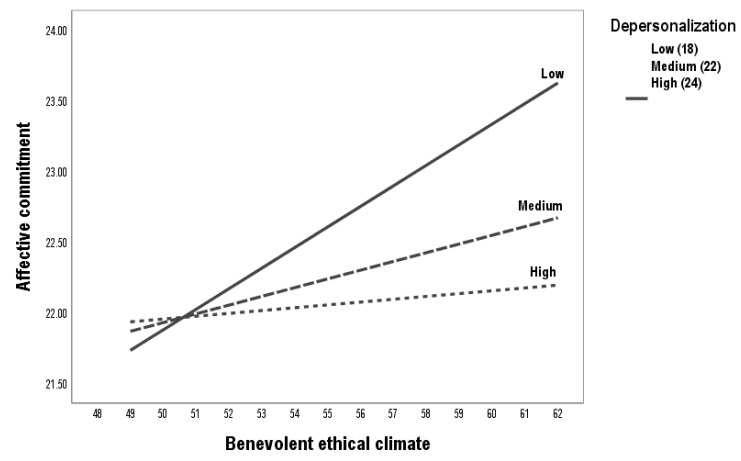
Moderation process between benevolent ethical climate and affective commitment.

**Table 1 ejihpe-15-00055-t001:** Means standard deviations and correlations for the main study variables.

Constructs	*N*	M	SD	BEC	F	GI	SR	DES	IM	AC
Benevolent ethical climate (BEC)	11	54.96	7.06	0.62						
D1: Friendship (F)	3	16.80	2.90	0.87 **	0.64					
D2: Group interest (GI)	4	21.20	3.10	0.92 **	0.72 **	0.61				
D3: Social responsibility (SR)	4	20.40	3.50	0.86 **	0.60 **	0.68 **	0.61			
Depersonalization (DES)	4	20.97	3.60	−0.19 **	−0.13 **	−0.15 **	−0.22 **	0.81		
Intrinsic motivation (IM)	5	27.08	3.05	0.36 **	0.26 **	0.31 **	0.39 **	−0.21 **	0.85	
Affective commitment (AC)	6	29.81	4.82	0.34 **	0.26 **	0.30 **	0.35 **	−0.57 **	0.29 **	0.83

Notes: *N* = 448, ** *p* < 0.01.

**Table 2 ejihpe-15-00055-t002:** Testing the mediating effect of benevolent ethical climate on affective commitment.

Predictors	Model 1 (AC)			Model 2 (IM)			Model 3 (AC)		
	*β*	*SE*	*t*	*β*	*SE*	*t*	*β*	*SE*	*t*
BEC	0.19 **	0.03	5.84 **	0.24 **	0.03	7.74 **	0.16 **	0.02	6.34 **
IM							0.31 **	0.07	4.16 **
*R* ^2^	0.25 **			0.21 **			0.23 **		
*F*	39.74 **			59.98 **			67.70 ***		
Indirect effect IM of BEC on AC: *β* = 0.05; SE = 0.02 [0.02; 0.08]									

Note: *N* = 448. Each column is a regression model that predicts the criterion at the top of the column. BEC = benevolent ethical climate; AC = affective commitment; IM = intrinsic motivation. ** *p* < 0.01, *** *p* < 0.001.

**Table 3 ejihpe-15-00055-t003:** Testing the moderated mediation.

Predictors	Model 1 (IM)			Model 2 (AC)			Model 3 (AC)		
	*β*	*SE*	*t*	*β*	*SE*	*t*	*β*	*SE*	*t*
BEC	0.26 **	0.09	3.94 **	0.52 **	0.16	3.18 **			
DES	−0.45 *	0.24	−1.87 *	−1.31 *	0.60	−2.17 *			
BEC × DES	−0.01 **	0.02	−3.08 **	−0.02 **	0.01	−2.90 **			
IM							0.20 **	0.47	3.43 **
IM × DES							−0.01 **	0.03	−2.24 **
*R* ^2^	0.16 **			0.15 **			0.15 **		
*F*	27.71 **			24.20 **			24.20 **		

Note: *N* = 448. EL = ethical leadership; TI = teleworking intensity; EE = emotional exhaustion; WA = work autonomy. * *p* < 0.05, ** *p* < 0.01.

## Data Availability

The original data presented in the study and the used questionnaire are openly available in The Open Science Framework repository at https://osf.io/w2g5b/?view_only=f8b9995262ed469eab5413f302dd83c4 (accessed on 10 February 2025).

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
