# Peer review of "Ethical Climate, Intrinsic Motivation, and Affective Commitment: The Impact of Depersonalization"

_ejihpe, 2025, doi:10.3390/ejihpe15040055_

Round 1

Reviewer 1 Report

Comments and Suggestions for Authors

This article aims to investigate the relationship between a benevolent ethical climate and affective commitment, taking into account the mediating effect of intrinsic motivation. At the same time, the study assesses the role of depersonalization as a factor to hinder the above-mentioned relationship.

Topic is very interesting and less presented in existing literature, as the authors mention in lines 109-111 “there is a limited amount of research that has analyzed the relationship between ethical climates and affective commitment” … with a limited focus on “samples that include public agency employees, higher education teachers, public accountants, and especially nurses” (lines 114-116) and without investigating the “underlying mechanisms involved in this relationship” (line 119). On another hand, existing research has not taken into consideration the potential mediating effect of intrinsic motivation or the moderating effect of burnout or depersonalization.

Recommendation: Introduction – lines 56-57, the other mention “To be more specific, we will focus on a particular ethical climate, which is benevolent…”. I would suggest to better define “benevolent climate” since it is a core term of the paper.

Contribution to theory and practice: This is clearly expressed by the authors the Discussion and Conclusion sections. On one hand the results show that a benevolent ethical climate is positively related to affective commitment which is consistent with previous studies. The novelty of this paper comes from the fact that intrinsic motivation mediates the positive relationship between a benevolent ethical climate and affective commitment and depersonalization plays a moderating role in the mediating mechanism.

Furthermore, the authors mention that “These results supported the argument that depersonalization is a risk factor for organizations in an era marked by digitization, excessive work climates, and excessive emphasis on organizational objectives rather than the care of interpersonal relationships” (lines 549-550).

The research method is complex and is well described. The sample consisted of 448 employees of the Colombian electrical sector. The economic context is well described in Chapter 2. The mediating effect was confirmed through a four-step method. The moderated mediation model was examined using SEM structural equations. The Conceptual Model is clearly described in Figure 1. The measurement of each variable and the three hypothesis are well defined and described.  The authors also mention the limitations in research (Chapter 5.2). The results of the structural equation model analyses are clearly following the hypothesis defined.

The authors are using relevant, updated literature references, pertaining to the investigated topic.

Reviewer 2 Report

Comments and Suggestions for Authors

This study examines   how a benevolent ethical climate impacts affective commitment in employees of the Colombian electrical sector, with intrinsic motivation acting as a mediator. The authors argue that a positive ethical climate enhances affective commitment, with intrinsic motivation strengthening this link whule  depersonalization diminishes the positive effects of both the ethical climate and intrinsic motivation on affective commitment.

The manuscript is well written and the authors employed  validated questionnaires and structural equation modeling.

I have some suggestions for revising this manuscript:

  1. To further strengthen the reliability analysis, I recommend that the authors clearly state the Cronbach's alpha coefficient or another appropriate index of internal consistency (such as composite reliability) for the scales used in the study. This would provide readers with a more detailed understanding of the reliability of the measurements and add to the overall transparency of the methodology.
  2. Introduciton: To enhance clarity for readers, it might be helpful to expand on how Self-Determination Theory (SDT) and Social Exchange Theory guide the development of the hypotheses. Providing more details on how these theories align with the study's variables could strengthen the theoretical framework and offer a clearer understanding for readers who may not be familiar with these concepts.
  3. Discussion, A greater emphasis on how depersonalization functions as a moderating factor and its unique contribution to the existing literature would underscore its significance. The practical implications could be further developed with more specific, actionable recommendations for enhancing the ethical climate and addressing depersonalization, particularly within the Colombian electrical sector or similar organizations. This would increase the practical value of the study.
  4. Limitations of this work: While some limitations are mentioned, the discussion could be strengthened by addressing potential biases inherent in the cross-sectional design and self-report method, and how these might impact the generalizability of the findings. Offering more specific suggestions for how these limitations could be mitigated in future research would provide additional clarity.
  5. Future research: it could be valuable to explore how different types of organizations or sectors (e.g., private vs. public) may respond differently to ethical climates. Additionally, examining the long-term effects of depersonalization on intrinsic motivation could guide further studies.

Round 2

Reviewer 2 Report

Comments and Suggestions for Authors

I am satisfied with the revisions and changes made by the authors, as they have adequately addressed my comments and suggestions. My only request is that the figure legends be placed below the figures.